# Characterizing the Rate of Spread of Large Wildfires in Emerging Fire Environments of Northwestern Europe using Visible Infrared Imaging Radiometer Suite Active Fire Data

Adrián Cardíl[1,2,3*], Mario Tapia[1,3*], Santiago Monedero[1], Tomas Quiñones[1], Kerryn Little[4], Cathelijne R. Stoof[5], Joaquín Ramirez[1], Sergio de-Miguel[2,3]

[1]Tecnosylva, S.L Parque Tecnológico de León, 24004 León, Spain
[2]Joint Research Unit CTFC - AGROTECNIO - CERCA, 25280 Solsona, Spain
[3]Department of Crop and Forest Sciences, University of Lleida, 25198 Lleida, Spain
[4]School of Geography, Earth and Environmental Sciences, University of Birmingham, Birmingham, UK
[5]Department of Environmental Sciences, Wageningen University, PO box 47, 6700 AA Wageningen, The Netherlands

*Correspondence to*: Adrián Cardil (acardil@tecnosylva.com)

* Both authors contributed equally

**Abstract.** In recent years fires of greater magnitude have been documented throughout northwest Europe. With several climate projections indicating future increases in fire activity in this temperate area, it is imperative to identify the status of fire in this region. This study unravels unknowns about the state of the fire regime in northwest Europe by characterizing one of the key aspects of fire behavior, the rate of spread (ROS). Using an innovative approach to cluster VIIRS hotspots into fire perimeter
isochrones to derive ROS, we identify the effects of land cover and season on the rate of spread of 102 landscape fires that occurred between 2012 and 2022. Results reveal significant differences between land cover types, and there is a clear peak of ROS and burned area in the months of March and April. Median ROS within these peak months is approximately 0.09 km/hr. during a 12-hour overpass, and 66% of the burned area occurs in this spring period. Heightened ROS and burned area values persist in the bordering months of February and May, suggesting that these months may present the extent of the main fire
season in northwest Europe. Accurate data on ROS among the represented land cover types as well as periods of peak activity are essential for determining periods of elevated fire risk, the effectiveness of available suppression techniques as well as appropriate mitigation strategies (land and fuel management).

## 1 Introduction

Wildfires are among the most common natural disturbances across the globe and play a key role in shaping many ecosystems.
In recent years fires have had an emerging impact in areas not traditionally considered fire prone such as the temperate region of northwest Europe. While large and severe fires in these regions were once considered an anomaly, in recent years the occurrence of fires of greater magnitude has been increasing (San-Miguel-Ayanz et al., 2021). In 2020, the Netherlands

experienced its potentially largest wildfire in recent history, affecting 710 hectares in the nature reserve of Deurnese Peel (Stoof et al., 2020). In the same vein, the United Kingdom had consecutive record fire seasons in 2018 and 2019 with burned areas of 18,032 ha and 29,152 ha, respectively, the largest burned area in the past 10 years (Belcher et al., 2021) and mostly affecting peatland areas. Several climate projections suggest increased fire activity in northwest Europe in the future due to projected drier and warmer weather (Krawchuk et al., 2009; Lung et al., 2013; de Rigo et al., 2017). Moritz et al., (2012) identified that temperate forests and grasslands are among the most vulnerable biomes in the mid-to-high latitudes to increases in the probability of wildfires, especially in the last three decades of the 21st century (2070–2099). Despite the projected elevated risk, fire behavior within this ecosystem is not well understood. Influential drivers such as vegetation and moisture conditions vary considerably from other parts of Europe such as the Mediterranean where fire and ignition conditions are better researched and understood.

Fire regimes are defined as long-term patterns of frequency, intensity, and fuel consumption of wildfires in a given area (Keeley et al., 2011). Anticipated changes in local climate are likely to drive alterations in fuels (types, structure, and heterogeneity) as well as conducive fire weather and thus fire occurrence and behavior. Temperate ecosystems, such as those in northwest Europe, which tend to have experienced limited fire exposure, may become increasingly fire-sensitive and susceptible to increases in fire frequency (McWethy et al., 2013; Kitzberger et al., 2016). Given that fire sensitivity is strongly dependent on interactions between vegetation phenology and fire seasonality, accounting for these factors is imperative when looking at fire behavior (Miller et al., 2019). Timing of phenological events is a major determinant of fuel availability and flammability and within temperate landscapes appears to be driven by changes in temperature (Fares et al., 2017; Chuine and Cour, 1999). The "green up" is likely an important phenological stage when it comes to fire behavior as this is when sap flow begins increasing vegetation moisture. Prior to this green up period, vegetation is drier and may be more susceptible to ignition as fuel moisture is among the most critical parameters affecting fire ignition and propagation (Parsons et al., 2016). *Calluna vulgaris* for example, a shrub commonly found in heathlands, is subject to extremely low fuel moisture levels in early spring as roots still frozen from winter dormancy can further limit water uptake alongside the phenological cycle, posing elevated ignition risks (Davies et al., 2010). Among the different aspects of fire behavior, the rate of spread (ROS) is a key indicator for characterizing fire regimes as it directly contributes to fire size and the overall residence time of the fire (Gill et al., 2008). Forest and fire management depend on accurate knowledge of fire behavior and, particularly, ROS for assessing appropriate fuel treatments (Vaillant et al., 2009). Furthermore, emergency responders such as firefighting operations strongly rely on accurate ROS data to determine their initial and extended attack as well as their fire suppression capabilities. Fires with lower spread rates are generally able to be attacked at the head using hand tools; however, those with higher spread rates will often be more intense and require special equipment such as dozers and retardant aircraft to be effective (Andrews, 2011). Accurate information on the ROS of fires can therefore help increase preparedness of land managers and emergency services across the region.

The relative novelty of fire in northwest Europe means that classifying a fire regime is rather difficult due to the scarcity of data and inconsistencies among record keeping (San-Miguel-Ayanz et al., 2021). European countries vary in their national classifications of fire, the quality of their fire-cause investigations, and the length of time of national databases (Tedim et al., 2015; Fernandez-Anez et al., 2021). Moreover, national fire databases tend to include data on fire occurrence and cause, but

not on fire behavior. Fortunately, remote sensing methods, through the use of satellites, provide spatially and temporally consistent data to permit further analysis of fire behavior, particularly ROS (Sá et al., 2017; Benali et al., 2016). Several studies have proposed methods to extract ROS from satellite imagery using various sensors, such as the Advanced Very High Resolution Radiometer (AVHRR) (Chuvieco and Martin, 1994; Al-Rawi et al., 2001). Several authors have turned to satellite thermal imaging from MODIS (Moderate Resolution Imaging Spectroradiometer) to reconstruct fire progression for larger

fires throughout the world (e.g., (Loboda and Csiszar, 2007; Veraverbeke et al., 2014; Jin et al., 2015; Chen et al., 2022)). Liu et al. (2018) employed the Himawari-8 geostationary satellite to extract the ROS in near real time from grasslands in Australia with promising results. The application of active fire detection products for identifying ROS has the potential to calibrate and validate fire spread models, and several studies have successfully demonstrated methodologies for rapid accurate assessments of ROS in near real time (Sá et al., 2017; Cardil et al., 2019). Many of the aforementioned methodologies and the sensors

implemented have been applied in regions where fires tend to be much larger than in temperate Europe, such as in the Mediterranean, California, or Australia (Andela et al., 2019; Chen et al., 2022). Our study relies on the Visible Infrared Imaging Radiometer Suite (VIIRS) satellite, which features a higher spatial resolution of 375 m for active fire detection (as opposed to 1 km for MODIS) and may capture more of the smaller fires prevalent in temperate regions.

The objective of this study was to characterize the ROS of large landscape fires within a temperate region facing an increasing risk of wildfires, namely northwestern Europe. We developed a new methodology to derive the ROS from the VIIRS active fire data product because of its global coverage, near real-time accessible data, and improved spatial and temporal resolution compared to MODIS (Schroeder et al., 2014). Subsequently, we assessed the effects of season and land cover type on ROS, as well as differences between countries. A thorough understanding of fires in these emerging regions of northwest Europe

will be imperative in developing functional fire management strategies and will serve as a baseline to assess future impacts of climate change.

## 2 Materials and Methods

### 2.1 Study area

For the purpose of this study, the boundaries of northwest Europe were defined by the northern Atlantic biogeographical region

above 49th parallel based on Sundseth et al., (2009), which includes many of the traditionally wet countries such as the United Kingdom, Ireland, the Netherlands, Belgium, Denmark, northern France and northwestern Germany (Fig. 1). We used the 49th parallel to delineate the boundaries of the study area to focus our analysis in the temperate region of northwest Europe, not

traditionally considered fire prone, instead of including northern Spain and southern France where fire regimes have been analyzed in previous research (Moreno and Chuvieco, 2013). The Atlantic biogeographical region is distinguished by an oceanic climate and occupies much of the flat lowlands along the Atlantic coastline. Overall, the climate is temperate, marked by its mild winters and cool summers with westerly winds and moderate rainfall throughout the year (Sundseth et al., 2009). The landscape tends to be intensely managed with considerable agricultural areas and expansive industrial and urban regions (Feranec et al., 2010). Nature environments are often isolated and discontinuous due to interwoven urban development spurred by dense populations. The represented vegetation types are diverse and feature heathlands, broadleaf beech forests, meadows, and peat bogs among many others. The climate in this region is ocean driven and as a result it is typically wet and humid for much of the year.

## 2.2 Visible Infrared Imaging Radiometer Suite (VIIRS) Data

We used data from the Visible Infrared Imaging Radiometer Suite that provides active fire data from the VIIRS sensor aboard the joint NASA/NOAA Suomi National Polar-orbiting Partnership (Suomi NPP) satellite launched in 2011. The VIIRS 375 m active fire product is described in Schroeder et al., (2014) and uses a multi-spectral algorithm to identify fire activity through 5 imagery channels (I-bands), 16 moderate resolution channels (M-bands) and a Day/Night Band (DNB). We used this product, which has been widely used in fire modeling applications, because of its higher spatial and temporal resolution and accurate response over fires of relatively small areas (Schroeder et al., 2014). Among VIIRS' greatest strengths is the ability to detect at moderate 375 m spatial resolution and global coverage approximately every 12 hours (Oliva and Schroeder, 2015); making it an ideal instrument for detecting the smaller fires we anticipated in our study area. VIIRS hotspot data were collected from the NASA Fire Information Resource Management System (FIRMS) portal (https://firms.modaps.eosdis.nasa.gov/) for the period of January 20th, 2012 to June 1st, 2022, as no earlier data was available.

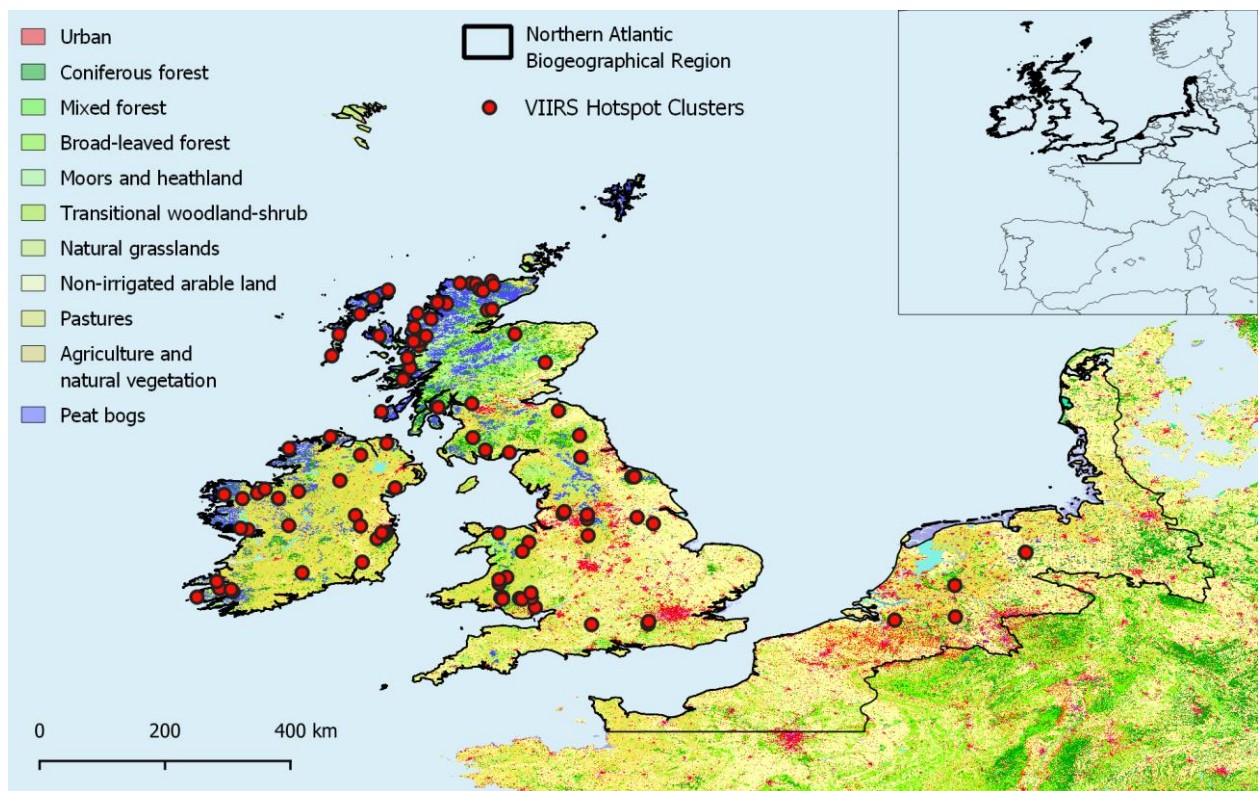

**Figure 1. Study area of northwest Europe, encompassing the United Kingdom, Ireland, the Netherlands, Belgium, Denmark, northern France, and northwest Germany within the northern Atlantic biogeographical region above 49th parallel (Sundseth et al., 2009). Locations of the fire hotspot clusters generated from VIIRS 375 m active fire product (VNP14IMGTDL_NRT; https://firms.modaps.eosdis.nasa.gov/download/) derived from 326,935 hotspot detections from January 2012 through June 2022. Land cover data source: Copernicus Land Monitoring Service's Corine Land Cover Map 2018 (2019a).**

## 2.2.1 Omission of Non-Landscape Fire Detections

VIIRS sensors detect fires through the identification of mid-infrared radiation emitted from heat sources across the landscape. As these hotspots may also include heat coming from other sources (such as gas flares at industrial sites), it was necessary to screen the VIIRS hotspots to remove thermal anomalies from sources other than landscape fire. As part of this process, we overlaid the polygons on the Copernicus Land Monitoring Service's Corine Land Cover Map 2018 ((2019a)) to distinguish landscape fires from other heat sources such as artifacts of heated plumes or a myriad of other anthropogenic phenomena. VIIRS hotspots identified within a 375 m boundary (resolution of VIIRS data) of land cover classified as urban or industrial were excluded from the database.

## 2.3 Clustering VIIRS data by fire incident based on space and time

We clustered VIIRS data by fire incident and satellite overpass, aiming to develop ROS vectors characterizing the fire growth. The clustering approach is a three-step process: 1) clustering in space, 2) clustering in time within each previous space cluster, and 3) filtering out clusters with less than 20 VIIRS hotspots, as these are insufficient to derive consistent ROS vectors. The clustering in space was carried out using a grid-growing clustering algorithm. All hotspots are projected into a 5 km cell size grid where clusters are defined as groups of interconnected cells in the grid (islands of cells containing hotspots). Cell size and 20 hotspot threshold values were heuristically set. To identify these clusters, we loop through the cells in the grid searching for an initial cell containing hotspots but with no assigned ID (not belonging to any cluster). The seed cell is assigned with an ID and then expanded (grown) among neighboring cells containing hotspots using a fast-marching method with 8 degrees of freedom and assigning the ID of the initial seed cell to all found cells. This process is done iteratively until all cells containing at least one hotspot have an ID. The method assures that any hotspot in the cluster has at least one neighboring point within $2\sqrt{2}$ the cell size distance of the grid. Then, the clustering in time was conducted by splitting the initial cluster into subclusters whenever there was a time elapse of 48 hour or more without a hotspot. This threshold value is heuristic and could be slightly modified without significant changes in the final result given that the fire frequency in the study area is usually low. The combined process of clustering by space and time leads to the final group of fire incidents used in the rest of the analysis.

## 2.4 Generating fire perimeters and ROS vectors

Once the hotspots were classified into individual fires, the next step was to generate fire perimeters. At each time step a Delaunay triangulation is constructed using the hotspots from that time step and previous ones. This triangulation method is closely related to Voronoi diagrams and has the important properties of not having intersecting edges, defining triangles with the nearest nodes, and reducing the number of possible interconnecting edges between points. This initial triangulation already contains the lines that will define the final fire perimeter, but to identify them in the triangulation we need to 1) apply an **α**-Shape algorithm, 2) identify the external edges of the mesh, and 3) construct a final geometry out of the individual edges. The **α**-Shape algorithm is applied to the mesh to remove those triangles whose circumradius (radius of circle circumscribed in a triangle) is greater than 10 km. This process splits the original mesh into independent triangulations if nodes (hotspots) are sufficiently distant from each other and creates holes inside the mesh if the node density is low. Basically, the value of **α** determines the maximum distance it is assumed hotspots could define a perimeter edge. In practice, **α** controls the "porosity" of the final shape since high values lead to a convex hull polygon while lower values increase the concavity of the perimeter. In this analysis the value of **α** was tested with different values (1, 3, 5, and 10 km), with 10 km being the optimal solution to create the fire spread polygons throughout the fire growth. The fire perimeters are now defined by the outer edges of the remaining mesh. These edges can be extracted by noticing that outer edges only belong to one triangle in the mesh while internal edges are shared between two triangles. Extracted edges are then ordered to form valid geometrical polylines and then aggregated together to form the final polygons representing the fire perimeters.

165      In this process the perimeter exactly connects the input hotspots without considering the actual VIIRS spatial resolution. This could be easily resolved by applying an external buffer to the perimeter equal to half the resolution distance of VIIRS (around 375 m). However, this does not affect the ROS calculation as the same procedure is used at each time step, and therefore the distance between consecutive perimeters is not affected.

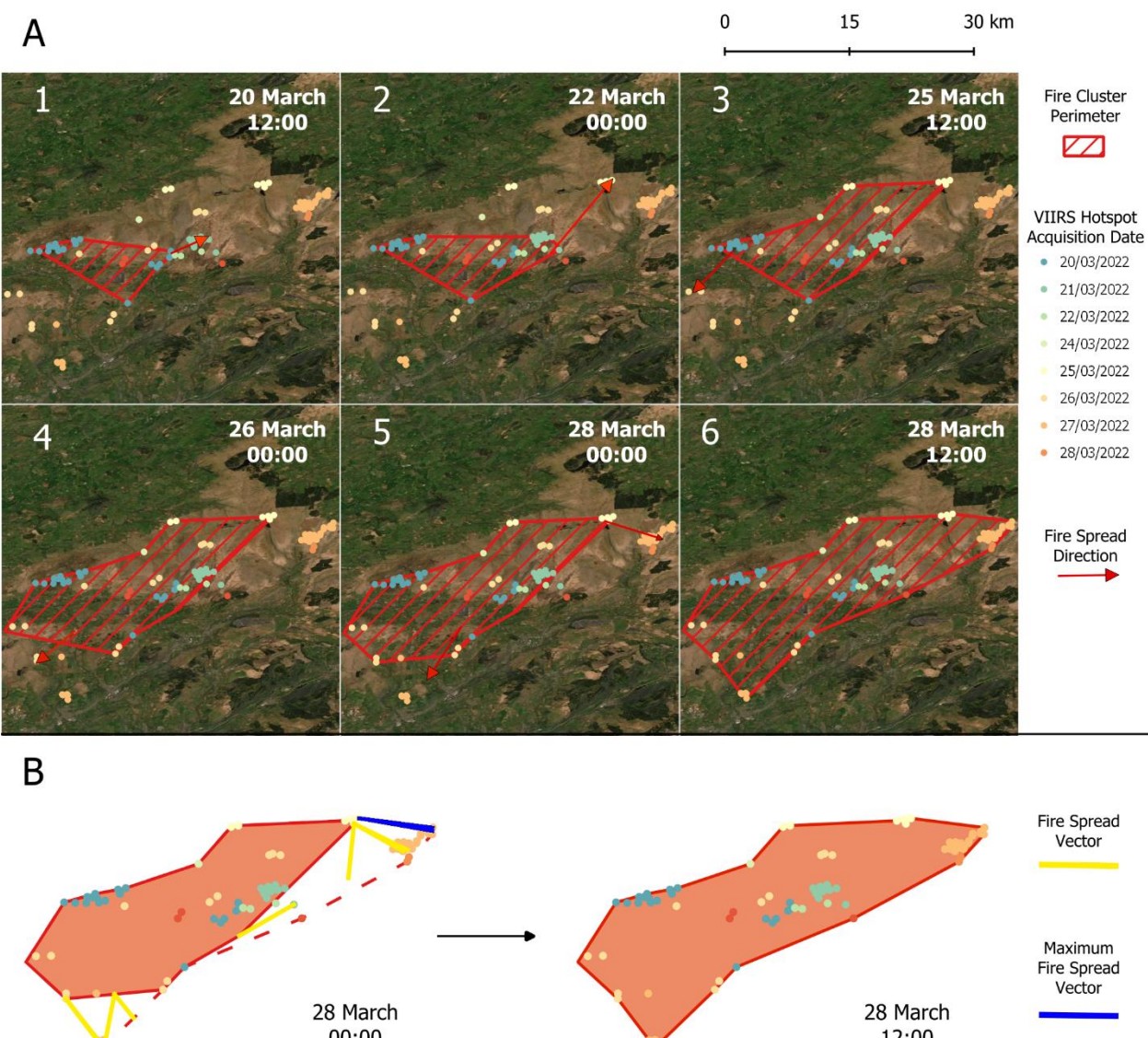

**Figure 2. Multipolygons and Rate of spread (ROS) vectors for the Mynydd Mawr fire in Wales (March 20, 2022), representing the**
170   **methodology used to delineate fire perimeters and Rate of Spread (ROS) vectors. A) Schematic of the fire growth in several time steps throughout the fire duration, delineated using the VIIRS hotspots (VIIRS 375 m active fire product at each satellite overpass (VNP14IMGTDL_NRT; https://firms.modaps.eosdis.nasa.gov/download/). Red arrows represent the main fire spread direction**

**schematically. B) Real ROS vectors (yellow lines) and perimeters generated (orange polygons) in two selected satellite overpasses. For each time step, the maximum ROS vector is selected for further analyses (blue lines), representing the ROS in the head of the fire.**

With the fire progression multipolygons developed, the rate of spread vectors could be calculated. For each vertex of the polygon at time t+1, the closest vertex of the parent polygon at time t was identified. Taking into consideration the distance and time between both points, we calculated the ROS of each spread vector (Fig. 2). To increase the accuracy of the spread vectors, the number of vertices at each polygon and time step was increased by adding one extra node between neighboring points. The approach developed is similar to that proposed by Hantson et al. (2022) but presents differences. Both clustering algorithms employ different parameters and threshold values to identify hotspot membership in a cluster. Our ROS generation process is based on node-to-node connections, which is a slightly more computationally efficient process, but could fall in an over- or underestimation of ROS for each fire. However, this is addressed also by the addition of the extra node, reducing error associated to this.

### 2.5 ROS Classification and Analytical Methods

Each individual fire contained many vectors for ROS at each vertex for each time step representing the fire progression in every direction (see an example in Fig. 2). To identify the widely utilized head of the fire, we grouped vectors by fire and timestep and filtered out the maximum ROS value for each fire and time step. As each time step also featured data on land cover, country of origin, and the month of incidence it became possible to draw comparisons between variables. We used a one-way ANOVA (Girden, 1992) and Tukey post-hoc statistical analysis to identify significant associations between ROS and land cover types. It is important to note that we performed this hypothesis test with only land cover as a factor, as there was an insufficient number of observations per group for the other factors. We also evaluated relationships between maximum ROS and burned area per fire through a linear model approach, transforming the variables as needed to meet the model requirements of homoscedasticity and linearity.

### 3 Results

### 3.1 Distribution of fire detections and classification of fire events

The 326,935 individual fire detections identified in our study area were filtered into 29,215 detections on wildland areas and clustered into 102 "real" landscape fires. Within these fires, we identified 327 ROS vectors that represented the ROS in the head of the fires for each satellite overpass throughout the fire growth and were used for further statistical analysis. The greatest number of fires were registered in the British Isles, while detections were scarce in continental Europe within the study area. In fact, there were no fires identified for the study period in Belgium, Denmark, or France. The final output for

each fire included timing polygons and burned area for each time step. For spread vectors, land cover and ROS vectors by time step throughout the fire duration were identified.


## 3.2 Burned Area

Peak burned area across the study region occurred in the months of March (40%) and April (26%), when 66% of the total burned area was observed (Fig. 3a). In April there was a greater percentage of fires (41%) compared to March (20%), indicating that while fires in April may be more frequent, they are also smaller. Therefore, there is a clearly defined peak fire season

from March and April. Outside of this fire season peak, fire activity starts increasing in February and decreasing from May and through summer (Fig. 3a). In terms of size, 54% of fires were smaller than 10 $km^2$. Fires exceeding 40 $km^2$ comprised 13% of the generated clusters (Fig. 3b). Mid-sized fires (from 10 to 40 $km^2$) comprised 33% of total wildfires.

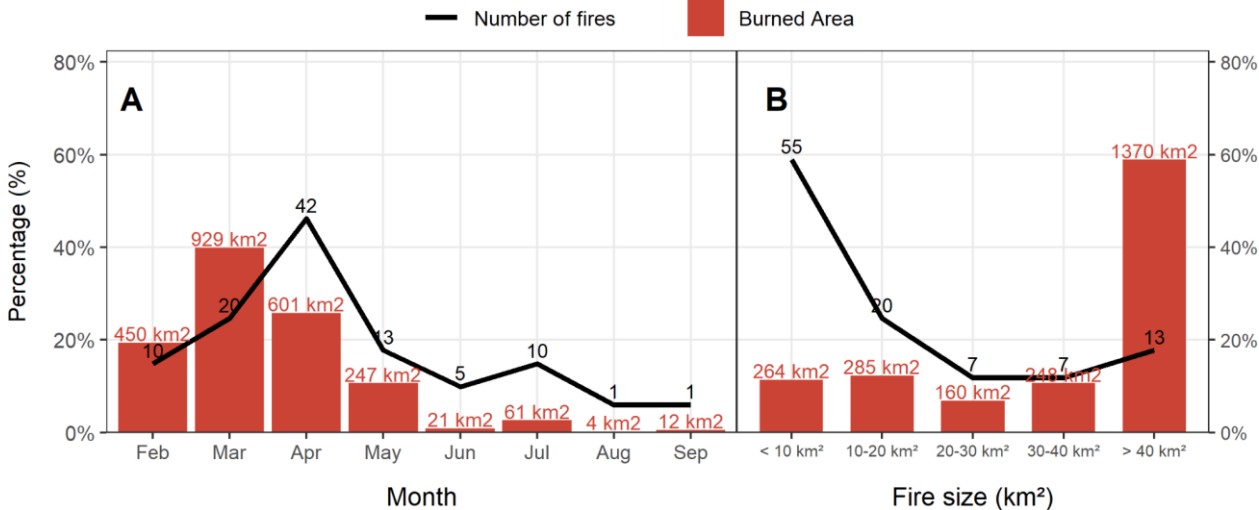

**Figure 3: Burned area percentages for northwest Europe calculated a) monthly and b) according to size distribution at the time the fire ended. Burned area may differ from that detected by other remotely sensed burned area products or delineated in the GIS systems maintained by fire managers because of the spatial resolution limitations of the active fire product used in this work (VNP14IMGTDL_NRT; https://firms.modaps.eosdis.nasa.gov/download/).**

## 3.3 Effect of Season on Rate of Spread

The months of February, March, and April witnessed the most active fire activity in northwest Europe (Fig. 4a). During this period median ROS was the greatest (0.13, 0.09, and 0.09 km/hr., respectively), and this was reflected in the greater burned area, exceeding 1,981$km^2$ (Fig. 3a). ROS from February to April also registered the highest ROS values. The remaining months

of June, July, August, September, and January featured a median ROS lower than 0.05 km/hr., and this also reflects the reduced

burned area (ca. 98 km$^2$) in these months (Fig. 4a). Mean ROS within northwest Europe was approximately 0.07 km/hr. for the entire study period, with half of our ROS observations falling within the 0.04 km/hr. to 0.14 km/hr. range (Fig. 4a). The data revealed that for the fires considered, 12-hour spread rates rarely exceed 0.34 km/hr.

At the country level, it is important to note that statements made for Germany and The Netherlands are based on few fires due to the scale of this research that focuses on larger landscape fires, which are relatively rare in these countries. Across northwest

Europe, landscape fires were detected from January to September, predominantly in February, March, and April (Fig. 4b). Peak ROS for England, Ireland, Scotland, and Wales occurred in late winter to spring months. Germany and The Netherlands only detected fires in 1–2 months. From the few records available, fires for Germany, The Netherlands, and Northern Ireland appear to deviate from this pattern of late winter–early spring fires. Germany reached its peak ROS in September, The Netherlands in April, and Northern Ireland in May.


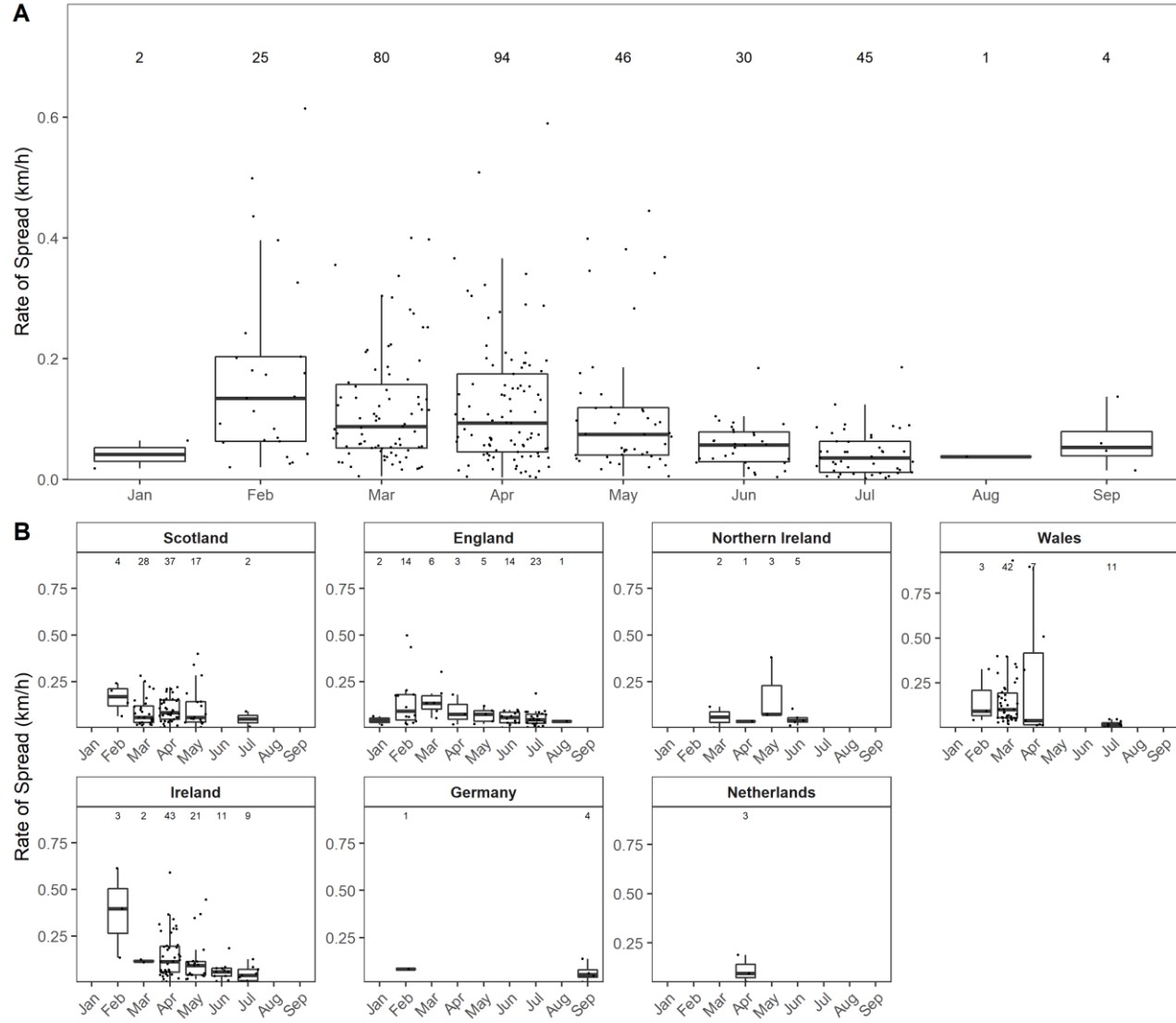

**Figure 4: Rate of spread for northwest Europe from 2012–2020 calculated monthly A) for the northwest European area (northern Atlantic biogeographical region) and B) per country. Points represent sample size, which is also indicated as numbers on top of each boxplot. Fires were not detected for months not represented in the figure.**


### 3.4 Land Cover Effects on Rate of Spread

The number of ROS vectors differed across land cover types, with the greatest number of ROS vectors recorded over peat bogs and moors and heathlands. Statistical analyses revealed significant effects of land cover on ROS (p=0.03; Fig. 5). Fire spread through coniferous forests was the greatest with a mean of 0.19 km/h., whereas fires spreading on transitional woodland-shrubs

had the lowest average ROS (0.04 km/h). The difference in ROS between these two types of land cover was significant (post-hoc analysis p=0.01). Natural grasslands, peat bogs, moors and heathland, and pastures cover types had averages ranging between 0.08 and 0.14 km/h. and no statistical differences were found between these cover types (Fig. 5).

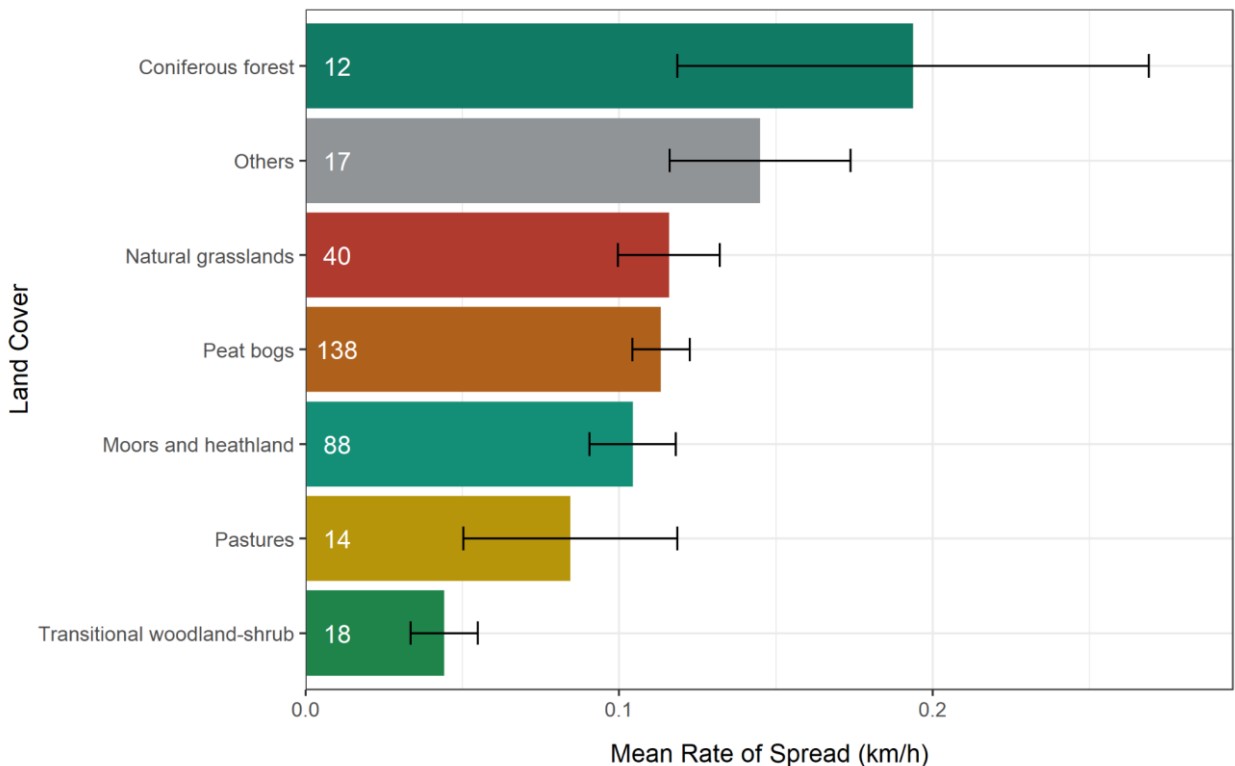

**Figure 5: Mean rate of spread and standard error across various represented land cover types in northwest Europe. The category "others" includes other land cover types with low representation (less than 10 maximum spread vectors by category). The number of ROS vectors for each land cover type is shown in the chart.**

### 3.5 Relationships between ROS and burned area

There was a significant positive linear relationship between maximum ROS and final burned area indicating that overall burned area increases with increasing of maximum ROS (Fig. 6). The model yields an $R^2$ value of 0.4.

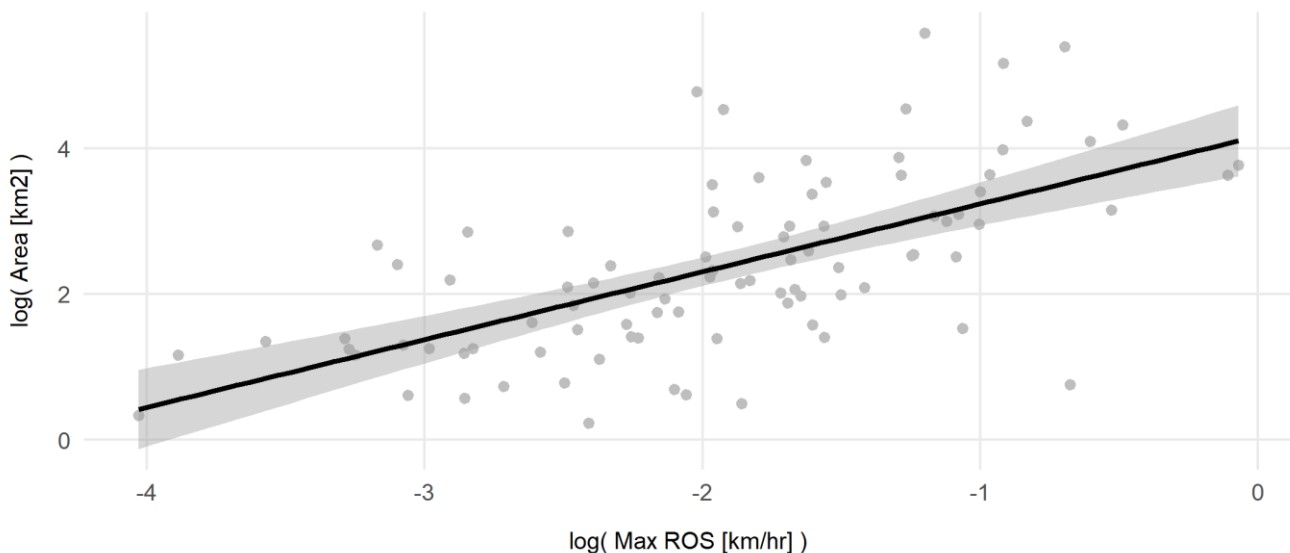

**Figure 6: Linear regression model for log-transformed variables (maximum ROS and burned area), indicating a positive relationship between maximum ROS and final burned area for each fire.**

## 4 Discussion

### 4.1 Burned Area & Peak Activity

The analysis of the intra-annual burned area is essential for determining periods of peak fire risk and impacts. Our results clearly indicate a major peak in burned area in late winter and spring (March and April) (Fig. 3a). Summer fires are unusual as they occur following the spring green up period when the vegetation moisture is typically high, hence large fires in temperate summer tend to require considerable arid conditions or drought. It is acknowledged that our data record of nine years occurs within longer-term fire activity and climate cycles that may influence the trends observed; however, these findings agree with research reporting fire seasonality using longer-term data sources (e.g., Belcher et al., 2021). Fires larger than 40 km$^2$ were responsible for most of the burned area, with equal contributions from fires in the 30-40 km$^2$, 20-30 km$^2$, 10-20 km$^2$ and <10 km$^2$ scales. The lack of fires smaller than 10 km$^2$ can likely be explained by the fact that the VIIRS satellite was unable to capture fires of this magnitude due to limitations of the temporal and spatial resolution  (Schroeder et al., 2014) and may also be because of the minimum threshold of 20 hotspot points to derive ROS vectors. Smaller fires are much more ubiquitous than larger fires as fire size distribution commonly follows the power-law function (Cui and Perera, 2008; Hantson et al., 2015). While it is likely that these smaller fires are more frequent than indicated, fires of this scale are less likely to be significant contributors to overall burned area (San-Miguel-Ayanz et al., 2021). It is important to remember that the fires included in this

study are of mid-to-large size and ROS estimates are thus representative of these fires. Smaller fires are likely to reduce the mean ROS, even though some small fires may have experienced high ROS over short time periods. Note that the smallest fire delineated and characterized in this work occurred in the Northumberland National Park (UK, 2018; latitude = 55.28°; longitude = -2.15°) accounting for 90 VIIRS hotspots, 5 timesteps and a burned area of 10 ha.

As mentioned above, the spatial distribution of fire clusters is concentrated in the UK and Ireland. This prevents statistical analyses between countries (but not between land covers) due to the low number of fires in other countries (Fig. 4b). At the same time, the low occurrence of mid-to-large size fires is also a description of the actual fire regime in this bioregion. The fires in mainland Europe within the study area are usually small and extinguished in a few hours (Stoof et al., 2020). Therefore, the number of hotspots and timesteps are limited, turning it difficult to accomplish the characterization of the ROS given that our method needs a minimum of two time steps. Our methodology targets the fires that are most likely to be challenging for management and may become more frequent with climate change.

Of the 102 fires considered in this study, it is unlikely that any were prescribed burns as most of the fires accounted for were outside the main prescribed burn window of October to March. Moreover, all of our fires burned overnight and prescribed burn codes in the study region do not permit burning overnight ((Heather and grass burning: rules and applying for a license, 2021; Guidance – The Muirburn Code, 2021; Department of Agriculture, Food and the Marine, 2021)). Furthermore, our clustering algorithm required a minimum of 20 hotspots to be identified as a potential fire, which is unlikely for a prescribed burn.

## 4.2 Spread Rates & Peak Activity

Spread rates were calculated for various countries throughout the year and for various land covers in a region where fire behavior data tend to be scarce with a limited history of records. The seasonal rate of spread analysis indicates peak burn area and ROS in the months of March and April (Fig. 3 & 4). The bordering month of February and May also featured heightened rate of spread and burned area values suggesting that February–May might present the extent of the peak fire season in northwest Europe. The fire season is likely longer than indicated within our study but due to sensor limitations (spatiotemporal resolution of data (Schroeder et al., 2014)) we are unable to capture and account for the smaller fires and can only ascertain periods of peak activity of the larger fires. While studies on fire regimes within this region are scarce, the timing and duration of the peak of fire season outlined in our study agrees with the fire season set out in various literature. De Jong et al. (2016) concluded that the majority of wildfire activity in the United Kingdom occurs from March to May (59% of events and 95% of the burn area). Likewise, the Irish Department of Agriculture, Food and Marine (DAFM) distinguishes the period of heightened fire risk from the months of March to June (Fire Management, 2021), and within The Netherlands, peak fire activity is typically in the spring and early summer (April to June) (San-Miguel-Ayanz et al., 2019).

The March–April spring fire season identified in northwest Europe is a clear contrast to the Mediterranean fire season, which features a minor peak in spring but a stronger peak towards the months of July–September, at the later end of summer (Pausas and Paula, 2012; Le Houérou, 1973). The Mediterranean fire season peaks in summer as the fuel moisture is lowest during this period within this climate (Chéret and Denux, 2007). However, this likely differs in northwest Europe where the temperate climate is typically wetter and more humid in summer, and the period of lowest fuel moisture tends to fall before the phenological green up period in late spring. Davies et al. (2010) reviewed temporal variations in moisture for *Calluna vulgaris* across Scotland and identified consistently low live fuel moisture contents in spring and consistently high moisture contents in summer, which were largely attributed to the flush of young green summer growth. Davies et al. 2010 also noted that vegetation moisture might be more sensitive to changes in weather conditions than seasonal trends, considering the rapidity of changes in fine fuels. Therefore, combinations of low nighttime temperatures, frozen ground, and relatively warm sunny days in the early spring may lead to elevated fire hazards. Significant correlations have been widely observed between wildfires and the phenological stages of vegetation across southern Europe (Moreira et al., 2009; Angelis et al., 2012), and it is likely that local phenology plays a role in the differences in peak fire activity in northwest Europe. The phenological cycle alters key characteristics of fuels, including biomass, spatial distribution, moisture content, and chemical composition, which are key determinants for fire behavior (Fares et al., 2017). Further research is needed to identify linkages between vegetative phenology, meteorological conditions, and fire occurrence in this region.

**4.3 Effect of Land Cover Type**

Our study did find a significant effect of land cover on ROS. Land cover has been shown to be among the main drivers of fire intensity and ROS through its influence on plant biomass, vegetation structure, and moisture content (Moreira et al., 2009). In a similar study, Loboda and Csiszar (2007) also reconstructed fire spread across Eurasia using MODIS and derived ROS and found significant effects of ecoregions. However, we recognize limitations in terms of satellite fire detection and resolution of vegetation type maps. Thus, future studies could benefit from the use of high-resolution fuel maps, which were not yet available when this research was conducted. Land cover may not be able to sufficiently encompass the complexity of fuel types present and, therefore, enhanced fuel type maps may allow further exploration of these relationships.

The ROS analysis among various land cover types found that ROS was fastest in coniferous forests. It was expected that coniferous-type vegetation had higher rates of spread, as it is documented that in Mediterranean regions this cover type has more fire-selectivity in heterogeneous landscapes, being qualified as one of the most fire-prone among vegetation types (Barros and Pereira 2014). In our study area, most coniferous forests were distributed in the British Isles and our proposed methodology identified several fire clusters associated with severe and complex fires such as Ceredigion Fire (~43 km$^2$ in 8 days March 2022), Galway Fire (~59 km$^2$ in 4 days May 2019), and Sliabh Beagh Fire (~38 km$^2$ in 2 days May 2017), in which there were

important coniferous vegetation areas that were burned. Moors and heathlands and peat bogs had the highest number of spread vectors in this study, though represented moderate ROS compared to coniferous forests. These two cover types included some of the more high-profile fires in recent years, such as the ~1,000 ha Saddleworth Moor fire (2018) and the ~16,000 ha Knockando fire (2019), as reported by The UK Forestry Commission (2019). Gazzard et al., (2016) performed an analysis of

wildfire incidents from the Incident Recording System data in the UK and identified that these moorland and bog fires account for 40% of the area burned in the region due to horizontal continuity of fuel, topography, and difficulty of suppression, which permits fires to spread. Consequently, this work points out how different land covers can reach similar values of burned area in the end, but at completely different ROS, having important implications for considering fire management across different land cover types.


Overall, ROS values were low to moderate. The fastest spreading fuel type, coniferous vegetation, only recorded a mean ROS of approximately 0.19 km/hr., which is not considered to be a fast-moving fire. The reason for such low values may lie within the methodology implemented, which produced average spread rates. The ROS vectors were derived from the VIIRS satellite, which has an overpass time of 12 hours. Developing the ROS vector over such a long-time frame is likely to underestimate

the actual ROS: it is unlikely that the fire progressed at a constant rate over the course of the timestep, especially as we are averaging the ROS over periods of night and day. Therefore, there is a degree of variation that should be taken when relying on these estimations from VIIRS. This may also be the reason why there appears to be less variation among the different land cover types.

**4.4 Relationships between ROS vectors magnitude and final burned area**

We expected to find a relationship between ROS and final burned area although there are many factors influencing the fire growth, including environmental variables and fire suppression actions. We found a significant positive relationship between maximum ROS and the final burned area. Peatland fires, the land cover with the highest number of observations in this study, account for one of the highest burned areas in the world (Rein and Huang, 2021), burning surface vegetation and underground

smoldering of peat soils with probably the slowest ROS among land cover types (Huang and Rein, 2017). The VIIRS satellite most likely only captured surface fires (not smoldering fires), which would represent the fires with the highest ROS values for this cover type. Hence, our findings point out the existence of a dynamic in which fast-spreading fires frequently end up in large burned areas in northwestern Europe.

**4.5 Implications for Fire Management**

Accurate data on ROS among the represented land cover types as well as periods of peak activity are essential for determining periods of elevated fire risk, the effectiveness of available suppression techniques, as well as appropriate mitigation strategies

(land and fuel management). While this knowledge is abundantly available for regions that are familiar with wildfires, emerging fire prone regions often lack research in part due to limited records and past fire exposure. northwest European forest

and fire managers may use these results to inform people that they should be prepared for incidents of wildfire in the months of March and April or even in late spring in years of severe drought (Fig. 3 & 4). These managers and fire suppression experts may also consider that the values for ROS are likely to be underestimated due to the methods implemented, particularly the averaging of ROS over 12-hour periods.

Knowledge of the underlying conditions permitting fire spread will be essential for determining appropriate suppression

methods. Furthermore, there is a need to account for the prevalent smaller fires that are not captured by VIIRS. Even fires of smaller magnitude have the potential for widespread impact due to a higher urban density throughout northwest Europe. Estimates for ROS of these smaller fires may require a more ground-based approach until there is development towards a monitoring system of satellite sensors with higher spatial and temporal resolution such as that of the Canadian WildFireSat suitable for real time emergency management (Johnston et al., 2020). Higher resolution information pertaining to fuels will be

necessary for identifying at risk vegetation types as well as for capturing fuel heterogeneity permitting fire spread. If increases in fire activity within this northwest European region are to be expected, it will be crucial to not only prepare for the fires experienced now, but those of more adverse behavior that may become more frequent in the future. Steps towards characterizing the current and future fire behavior in this region should be further addressed to ensure future risks can be properly approached and mitigated.


**5 Conclusions**

The findings obtained in this study provide among the first estimations of wildfire ROS for northwest Europe and present a novel methodology towards the characterization of fire behavior in this newly fire prone region through the use of satellite hotspot detection. This technique identified 102 fires and has proven to be a useful alternative in the absence of field-based

measurements and years of data collection. This allowed us to characterize the ROS on different land cover types in northwest Europe and identify peak months of fire activity, providing key information for efficient fire management. It was also possible to establish a relationship between the ROS and the final area burned in this area of Europe, characterized by different environmental conditions compared to studies developed in southern Europe. Steps towards characterizing the fire behavior in this region should be further addressed to ensure future risks can be properly approached and mitigated. Future analyses to

characterize fire behavior and fire regime in northwest Europe should consider the fire frequency, duration, as well as the meteorological conditions contributing to ignitions.

*Data availability*. The data collected and produced is available on the Zenodo platform. The DOI and link of access is
https://doi.org/10.5281/zenodo.6330201

*Author Contributions.* M.T. and A.C. conceived the idea; S.M. developed the algorithm and ran the analyses; M.T., T.Q. and A.C. ran subsequent analyses and interpretation; M.T. led the writing of the manuscript; A.C. and T.Q. led the review of the manuscript after reviewers' comments and suggestions. All authors contributed critically to writing and editing the drafts and gave the final approval for publication.

*Competing Interests.* The contact author has declared that neither they nor their co-authors have any competing interests.

*Acknowledgements.* This project has received funding from the European Union's Horizon 2020 research and innovation programme under the Marie Skłodowska-Curie grant agreement No 860787 (PyroLife Innovative Training Network
https://pyrolife.lessonsonfire.eu/), a project in which a new generation of experts is trained in integrated fire management.

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
