# Peer review of "Characterizing the Rate of Spread of Large Wildfires in Emerging Fire Environments of Northwestern Europe using Visible Infrared Imaging Radiometer Suite Active Fire Data"

_Natural Hazards and Earth System Sciences, 2022_

## Author Response (AR1)

**Reviewer 1**

Tapia et al. presented a new approach to cluster VIIRS hotspots and derive the rate of spread (ROS) for each fire in this manuscript. They applied this approach to landscape fires in northwestern Europe and examined the relationship between ROS and land cover type, season, as well as geographical locations (countries). The ROS is closely related to other fire behavior and the impact of fires on ecosystems. So the documentation of ROS across northwestern Europe in this manuscript provided a good reference for future studies. However, there are some issues in the current manuscript that prevent me from recommending it to be accepted by NHESS.

1. Need more complete description of the VIIRS data and the approach.

For VIIRS fire data, the authors need to provide some background information on the satellite, the remote sensor, the data product (including the name, resolution, uncertainty, etc.), as well as the data filtering approaches. For example, what exact fire product did you use, the monthly data or the near real time (NRT) data? Is the resolution 375 m for all the pixels?

**We have added more details about the VIIRS product in section 2.2. "We used data from the Visible Infrared Imaging Radiometer Suite that provides active fire data from the VIIRS sensor aboard the joint NASA/NOAA Suomi National Polar-orbiting Partnership (Suomi NPP) satellite launched in 2011. The VIIRS 375 m active fire product is described in Schroeder et al., (2014) and uses a multi-spectral algorithm to identify fire activity through 5 imagery channels (I-bands), 16 moderate resolution channels (M-bands) and a Day/Night Band (DNB). Specifically, we used the VNP14IMGTDL_NRT near-real time product which has been widely used in fire modeling applications, in part because of its higher spatial and temporal resolution and accurate response over fires of relatively small areas (Schroeder et al., 2014). Among VIIRS's greatest strengths is the ability to detect at moderate 375 m spatial resolution and provides global coverage approximately every 12 hours (Oliva and Schroeder, 2015); making it an ideal instrument for detecting the smaller fires we anticipated in our study area. VIIRS hotspot data were collected from the NASA Fire Information Resource Management System (FIRMS) portal (https://firms.modaps.eosdis.nasa.gov/) for the period of January 20th, 2012 to June 1st, 2020 as no earlier data was available."**

Some of the information on the fire clustering algorithm is also missing or not clear. How was the temporal grouping performed? How were the timing and the land cover of a fire determined? How did you verify whether a fire is real fire? See the 'Minor comments' below for detailed questions.

**We have added more details about both the temporal and spatial clustering we performed. Also, we have addressed all the minor comments raised by the reviewer below.**

**"The clustering in space was carried out using a grid-growing clustering algorithm. All hotspots are projected into a 5km cell size grid where clusters are defined as groups of interconnected cells in the grid (islands of cells containing hotspots). To identify these clusters we loop through the cells in the grid searching for an initial cell containing hotspots but with no assigned ID (not belonging to any cluster). The seed cell is assigned with an ID and then expanded (grown) among neighboring cells containing hotspots using a fast-marching method with 8 degrees of freedom, and assigning all found cells the ID of the initial seed cell. This process is done iteratively until all cells containing at least one hotspot have an ID. The method assures that any hotspot in the cluster has at least one neighboring point within $2\sqrt{2}$ the cell size distance of the grid."**

**"The clustering in time was conducted by splitting the initial cluster into subclusters whenever there was a time elapse of 48 hour or more without a hotspot. This threshold value is heuristic and could be slightly modified without significant changes in the final result. The combined process of clustering by space and time leads to the final group of fire incidents used in the rest of the analysis."**

The approach to calculate ROS, which is the centerpiece of this study, also lacks important information. Sometimes the descriptions are contradictory or confusing (e.g. Which ROS statistics did you use? You mentioned maximum ROS, median ROS, and average ROS in the manuscript). The interpolation algorithm of vertices is also unclear to me.

**For statistical differences of ROS among land cover classes we used Mean ROS (Fig. 5). Graphical representation of ROS variation among months we used the median, as they were schematized as boxplots (Fig. 4)**

**We addressed this and edited the methodology section for a further explanation concerning about vertices and vector generation:**

**"With the fire progression multipolygons developed, the rate of spread vectors could be calculated. For each vertex of the polygon at time t+1 the closest vertex of the parent polygon at time t was identified. Taking into consideration the distance and time between both points, we calculated the ROS of each spread vector (Fig. 2). To increase the accuracy of the spread vectors, the number of vertices at each polygon and time step was increased by adding one extra node between neighboring points."**

2. Concerns about methodology

 The VIIRS active fire data represent the center location of each pixel. The pixel size is ~375m at nadir and varies with scan angle (can be much bigger at the edge). The current fire shape algorithm used these center locations directly, without considering the detection uncertainty of the data. Can this influence the calculated ROS? In the spatial grouping of fire pixels, 5km rasters were used. This leads to possible merging of fire pixels with distances at a maximum of ~14km. Is this too conservative? How does the variation in this value affect the clustering and ROS? For the Alpha value, why did you use 1km? Should you estimate the uncertainty related to this value?

**We addressed this and edited the methodology section for a further explanation of this process:**

**"Basically, the value of α determines the maximum distance it is assumed hotspots could define a perimeter edge. In practice, α controls the "porosity" of the final shape since high values lead to a convex hull polygon while lower values increase the concavity of the perimeter. In this analysis the value of α was tested with different values (1, 3, 5 and 10 km), being 10 km the optimal solution to create the fire spread polygons throughout the fire growth. The fire perimeters are now defined by the outer edges of the remaining mesh. These edges can be extracted by noticing that outer edges only belong to one triangle in the mesh while internal edges are shared between two triangles. Extracted edges are then ordered to form valid geometrical polylines and then aggregated together to form the final polygons representing the fire perimeters.**

**Notice that in this process the perimeter exactly connects the input hotspots without considering  the actual VIIRS spatial resolution. This could be easily  fixed by applying an external buffer to the perimeter equal to half the resolution distance of VIIRS (around 375 m). In any case, it is important to notice that this does not affect the ROS calculation since the same procedure is used at each time step, and therefore the distance between consecutive perimeters is not affected."**

3. Statistical robustness

The other issue I'm concerned about is whether some of the analyses (and results) are statistically robust. The total number of fires you used for analysis in this manuscript is only 254. While this number may be sufficient for whole-regional statistical analysis, you further divided the fires into different seasons, land cover types, and countries. I doubt the sample number is enough for all the categories.

**Sample size relates to every time step within individual fires (e.g. A single fire that has 4 time steps must have the same number of maximum spread vectors) Hence, the sample size is greater than the number of fires. It is important to note that statistical analysis was carried out just for cover types.**

**The analysis was redone. Now we have 102 fires with a total number of 327 vectors. Figures were edited and now all indicate the sample size for each cover/month. For the statistical analysis, land covers with a sample size lower than 10 were excluded and reclassified as "Others" (Figure 5).**

4. More analysis on ROS

It's good to see the ROS variations across different land covers and seasons. But I expect the authors to do more analysis to support the usefulness of the dataset. Some examples include, but not limited to, the relationships between burned area increment and ROS; the influences of weather variables on ROS; the statistical relationship between ROS and fire size.

**New hypothesis and post-hoc test was carried out. We did a new analysis to assess the relationships between burned area and ROS as suggested.**

Minor comments;

Line 23: "suggesting that may present the extent of the fire season"

What does the 'that' refer to? May change to something like 'this period' or 'these months'.

**Addressed. We have used "these months".**

Line 38: "Moritz et al. ((Moritz et al., 2012)". Some citations (such as this example) are not formatted correctly.

**Done.**

Lines 39-40: "in the last quarter of the 21st century (2070–2099)". 2070–2099 has 30 years and is more than a quarter of a century.

**This is true. Addressed.**

Lines 117-118: "its higher spatial and temporal resolution compared to other satellites such as MODIS". Some satellites have higher spatial or temporal resolution than VIIRS. Need additional defining words for 'other satellites'

**We agree with the reviewer. We have changed the sentence.**

Lines 134: "VIIRS detections are points scattered in time and space"

This is not quite true. A location record in the VIIRS fire data file does not represent the exact burning location, which could be anywhere within a pixel of the VIIRS footprint (which also varies with scanning angle).

**We agree with the reviewer. We have removed this sentence from the manuscript.**

Line 143-145: "The clustering in time was conducted by ordering the space clusters by time and creating divisions or break points if there was a time difference greater than 48 hours in between consecutive points."

The clustering method at the temporal axis is not clear for me. How did you determine which space clusters should be tested temporally? What do 'consecutive points' mean? Individual fire center locations, or the 5km pixels?

**Thanks for the commentary. We have edited the methodology section and further explain this process:**

**"...clustering in time was conducted by splitting the initial cluster into subclusters whenever there was a time elapse of 48 hour or more without a hotspot. This threshold value is heuristic and could be slightly modified without significant changes in the final result given that the fire frequently in the study area is usually low. The combined process of clustering by space and time leads to the final group of fire incidents used in the rest of the analysis."**

**Clusters were tested spatially for different alpha values. The optimal solution for this was to establish a 10 km value. Thereafter, temporal clusters were generated from spatial clusters. Consecutive points are considered centered-points data.**

Line 183-184: "Copernicus Land Monitoring Service's Corine Land Cover Map 2018 ((2019a)) to distinguish landscape fires from other heat sources such as active volcanoes, artifacts of heated plumes". Can the Copernicus Land Cover Map be used to distinguish volcanic eruptions from fires?

**We have changed the sentence. Copernicus Land cover Map was used to remove hotspots within urban/industrial areas and not for volcanic eruptions. Hotspots from volcanoes are not relevant in our study area.**

Line 192: "As each timestep also featured data on land cover"

What land cover product did you use for this purpose? Still Copernicus Land Monitoring Service's Corine Land Cover Map 2018? Please also describe how you determined the land cover type for each fire when the fire is big enough to cover different land cover pixels.

**That sentence was related to analyzes on each vector, not for individual fires/polygons. When we evaluated ROS variation over land covers, it was at the vector level and the most abundant land cover type passing through each.**

Line 198: "ANOVA and Tukey statistical analysis"

This statistical method may not be familiar to many readers. Please add a reference.

**Reference is added now**

Line 204-205: "of which 254 were verified to be "real" landscape fires"

Please specify the details about the way you verified the real fires.

**The filtering was carried out for VIIRS hotspot with a buffer of 375m. We removed those hotspots that intersected with the urban/industrial areas of the CORINE Land Cover.**

Line 209: "timing of the fire, the burnt area, the land cover, and the maximum ROS"

For a fire covering multiple raster pixels and time steps, how did you determine the 'timing of the fire' and 'the land cover' for the whole fire?

**Algorithm clusterization process outputs several timestep-polygons for each fire and every timestep has its own time footprint. Each timestep generated is a clusterization of near-time pixels.**

**As mentioned above, land cover analyses were carried out at a vector level, not for entire polygons.**

Line 219-221 : "On the other hand, fires less than 0.01 km2 were rarely detected with our satellite-based analysis, comprising approximately 0.002% of the total burned area and 1% the total number of fires. Fires between 0.01–0.1 km2 were also seldom observed with 0.3% of the burnt area set by 10.2% of total fires."

In the method section (Line 136), you mentioned you "filtering out clusters with less than 20 VIIRS hotspots". This filtering will reduce the number of small fires (in <0.01km2 and 0.01-0.1km2 bins) for certain. So the fraction of the number of fires in different size groups can be artificial.

**Thanks for the commentary, we have addressed it in the re-edited version. We agree with the reviewer, we discuss this since new improvements in technology may allow enhancing the recognition and ROS characterization of small fires.**

Line 229: "It was during this period that the median ROS was the greatest"

In the Method section (Line 192), you said you used 'maximum ROS', here you said you used 'median ROS'. Did you calculate the maximum ROS for a single fire (at a single time step, or for all time steps?), and then calculate the median ROS from all fires? The description needs to be clear in the Methods section.

**We only used the mean and median ROS to discuss our results. Note that "maximum ROS" is not used as a statistical metric. Each fire may have different time steps during the fire growth and each time step has different ROS vectos. We selected the vector with the highest ROS to consider the spread of the head of the fire.**

Line 278-280: "The lack of fires smaller than 1 km2 can likely be explained by the fact that the VIIRS satellite was unable to capture fires of this magnitude due to limitations of the temporal and spatial resolution."

Again, is this because you filtered fires with less than 20 VIIRS hotspots?

**More than 20 hotspot points are needed to derive consistent vectors. Hence, we are grateful for your commentary as we added this limitation in the sentence you indicated.**

Line 330: "our study did not yield any significant effect of land cover on ROS"

This conclusion is not consistent with that shown in Figure 5, where we can see the obvious differences in the RoS for different land cover types.

**We agree with the reviewer. The new analysis shows statistically significant differences among land cover types.**

Line 369: "lie within the methodology implemented, which produced average spread rates. "

Now you say 'average spread rates'. So it's not the 'maximum ROS' you mentioned in line 192?

**This was mentioned for every raw ROS vectores generated, before the maximum was selected for each time step for further analysis, as we mentioned above.**

Figure 1. The caption says "b) VIIRS hotspots retrieved from the area of interest". But I didn't see hotspots in this panel. I only see land cover types shown on the map.

**Thank you for this comment. It was a mistake that we have addressed.**

Figure 4. Considering there are only 254 fires in total, the number of samples in each country-month bin is expected to be small (It's also good to show this number in the Figure). The statistical robustness needs to be addressed.

**Now we have a total number of 102 fires and 327 vectors. For every new figure presented in the new version, the number of observations for each category is indicated.**

Figure 5. What are the 'n' values referred to? I don't think they are numbers of fires, since the total is way above 254 (the total fire number).

**Every fire has several time steps. Every time step has several ROS vectors and we selected the vector with the highest ROS by time step. The number of observations (n) is related to the number of maximum ROS values for each time step. In this new version, 102 fires were clusterized with 327 maximum vectors values.**

**Reviewer 2**

The preprint "Characterizing the Rate of Spread of Wildfires in Emerging Fire Environments of Northwestern Europe" by Mario Tapia et al. presents a systematic investigation of wildfire rate of spread (ROS) derived from VIIRS 375 m fire products. The article is well-presented and overall clearly written.

The authors propose a methodology to formalize the quantification of a fire behavior variable which, undoubtedly, is frequently estimated in an ad-hoc manner by users of fire detection data, specifically from the fire and natural hazard community, when managing a specific fire event. The authors' approach is potentially suitable to use as the basis for developing a remotely sensed product of interest to the user community. As such, the work is innovative and of undeniable interest. As the article's area of interest is north-western Europe - not a region known for very large or disruptive wildfires and therefore, in the light of climate change induced greater expected future prevalence of the wildfire hazard - it also contributes to enhanced understanding of the fire regimes in this part of the world.

Notwithstanding these strengths in scientific significance and quality of the presentation, I perceive a certain number of weaknesses that should be addressed before the manuscript is accepted for publication.

Definition and structure of the study area (2.1). To the reader who is not immersed into the study of this area, the choice of study area appears at least somewhat arbitrary. Was the intent here to study an area of Europe somewhat under-represented in the study of wildfire, and therefore to apply boundaries so as to stay clear of the Mediterranean region in the south and the Scandinavian/boreal region in the west and north? The eastern boundary and the choice of the 49th parallel should be better justified. If this is a commonly studied area thus delineated, a citation should be added. This point may appear as a formality, but I believe it is more significant than that, especially when it comes to the statistics presented for the countries outside the British Isles. For example, a quick look into German fire statistics shows that, contrary to the findings presented here, wildfire activity tends to peak in the month of August. It also shows, however, that German wildfires are dominated by fire events in the Land of Brandenburg, which is cut in half by the eastern border of the study area here. Given this kind of limitation, and the extremely small sample size of fires outside the British Isles, I do not think that per-country statistics (3.3 and Fig 4) should be presented for the countries other than the British Isles.

**We have further clarified the delineation of the study area. Please, see section 2.1 and Figure 1. "For the purpose of this study, the boundaries of northwest Europe were defined by the northern Atlantic biogeographical region above 49th parallel based on Sundseth et al., (2009), which includes many of the traditionally wet countries such as the United Kingdom, Ireland, the Netherlands, Belgium, and Denmark, northern France and northwestern Deutschland (Fig. 1). We used the 49th parallel to delineate the boundaries of the study area to focus our analysis in**

the temperate region of northwest Europe, not traditionally considered fire prone, instead of including northern Spain and southern France where fire regimes have been analyzed in previous research (Moreno and Chuvieco, 2013).”

We understand the concerns of reviewer 2 regarding the fire activity in Germany since she/he is considering the Land of Brandenburg which is out of the study area (northern Atlantic biogeographical region), probably with a different fire regime.

We do think it is worthwhile to keep Fig 4 as it is. In this sense, the readers can better understand that the fire activity in these regions is low and when the largest fires were identified. We have also improved the caption of Fig 4 to address the reviewer's comment.

VIIRS data description, limitations, pre-processing and exploratory statistics.  Section 2.2 needs to clearly describe which VIIRS product was used (I presume VNP14IMGTDL_NRT), and also confirm that the study is based only on S-NPP VIIRS data  (no NOAA-20 data, which would duplicate the data record in the last year or so).

We used the VNP14IMGTDL_NRT product based only on S-NPP VIIRS data. We have clarified this in the manuscript and added more details on the VIIRS data description.

Given that the filtering for retained fire detections ("real" fires) ended up rejecting ~90% of fire clusters, it is odd that clustering happened before filtering. The filtering criteria are also not very clear. A cleaner approach would have been to filter by land cover type (or, potentially, by using available GIS data of nature preserves, forested areas etc.) firstand then cluster the remaining events. Regardless, it would be instructive to see some minimal exploratory statistical description of the retained fire events - how many by year? By land cover type? Their final number - 256 - is very small compared to the known fires in this area over the 9 years of the study time. This is to be expected as it is known that VIIRS  misses many detections. But this fact is a rather relevant limitation of the study, which needs to be discussed. As-is, it seems likely that the results are dominated by particularly large fire years in specific sub-areas, which may very well skew the ROS statistics presented in the results. For example, the 2019 peatland fires in Scotland and Northern England may account for a rather outsized part of the results.

The reviewer is right. After checking the data processing we realized that, in fact, we filtered the VIIRS hotspots before the clustering process. To further clarify this, we have put the old section 2.5 in the current section 2.2 named "Visible Infrared Imaging Radiometer Suite (VIIRS) Data".

**We recognize that more fires occurred in the study period in the region of interest. This was discussed in section 4.1 as follows "The lack of fires smaller than 1 km2 can likely be explained by the fact that the VIIRS satellite was unable to capture fires of this magnitude due to limitations of the temporal and spatial resolution.".**

**While it is likely that these smaller fires are more frequent than indicated, fires of this scale are less likely to be significant contributors to overall burned area (San-Miguel-Ayanz et al., 2021). Moving forward, it is important to consider that the fires included in this study are of mid to large size rather than smaller fires especially when it comes to estimating the ROS as smaller fires are likely to reduce the mean ROS, despite the fact that some small fires may have experienced high ROS over short time periods. Therefore, we can say we studied the ROS of the largest fires that usually lead to the highest ROS.**

Algorithm description. In my view, the chief interest of this work is the ROS vector generation algorithm. More effort should be deployed to describe its strengths and limitations. For example, in section 2.4 and Fig. 2, the fire detections are not points, but VIIRS pixels of at least a size 375 x 375 m (or substantially larger if the acquisition is off-nadir). The VIIRS data includes complementary information (which may include x and y pixel extent, depending on the product used, and does include a confidence rating) - was this information used in any way and how stable are ROS derivations to this. Also, fire spread has an extremely strong diurnal pattern, so the reporting of spread km/h is a value that has undergone averaging. In Fig 2 you present an example with ~14 h between successive acquisitions, but VIIRS overpasses can re-image the same spot with an interval of 90 min or up to several days, and the ROS values you would obtain would be radically different given the diurnal variation. At the very least you should report the distribution of delta-t values used for ROS calculations, and possibly apply a correction factor based on expected temporal fire activity patterns.

**Fire detections were approached as points (hotpots from VIIRS active fire data products). Limitations and assumptions of this approach were mentioned throughout the manuscript. We can only produce average spread rates as we have a 12~14 hours satellite overpass. However, we recognize that temporal variations of ROS may occur during this time period, especially considering diurnal variation as you mentioned.**

Some more localized comments:

16/17: Given the substantial statistical limitations of the study, I think that this sentence overstates the amount of insight gained for understanding of fire regimes.

**We have removed the word "important" here. We do think we are unraveling unknowns about the state of the wildfire regime characterizing the rate of spread.**

31: An anomaly is probably an understatement. There is a long record of fire use for lanscape management by successive human populations.

**We agree with the reviewer. We have refined this sentence: "While large and severe fires in these regions were once considered an anomaly, in recent years the occurrence of fires of greater magnitude has been increasing (San-Miguel-Ayanz et al., 2021).".**

34-36: The increasing peatland megafires should probably be mentioned here, especially since my suspicion is that they dominate the dataset this study is based on.

**We agree with the reviewer. "In the same vein, the United Kingdom had consecutive record fire seasons in 2018 and 2019 with burnt areas of 18,032 ha and 29,152 ha, respectively, the largest burn area in the past 10 years (Belcher et al., 2021), most of them affecting peatland areas. ".**

39: Moritz et al. (2015) reference needs to be re-formatted.

**Done.**

90: The capitalization of n/Northwest/ern Europe should be unified.

**Addressed. "northwest/ern Europe".**

120: Extraneous semicolon.

**Addressed.**

Figure 1: Please revise the legend. It should indicate the origin of the land cover classification. Also, the hotspots - or hotspot clusters? - are in subfigure a), not b).

**Thank you for this comment. The figure caption was wrong. We think it is fine now after adding the data sources.**

101/102: The capitalization choice "northern Atlantic Biogeographical region" is odd here and in the following.

**We have used "northern Atlantic biogeographical region"**

137-143 [re: spatial clustering] There are other algorithms that also do not require cluster centers and number of clusters to be indicated a priori. With about 40,000 detections per year this is not a data volume that would be a problem for example for a variant of DBSCAN. Not that the outcome is going to be very different, but the clustering methodology comes across as somewhat clunky. The 5 km and 20 detections threshold aren't very well justified. (Also, where these distances measured in a projected coordinate system, that is, was the whole dataset reprojected, and if yes to which coordinate system?) Later, in the Results section, there is insufficient reporting on the impact of the parameter choice in clustering on the final dataset of fire events.

**The number of annual detections without considering urban/industrial areas was lower than 40,000. Especifically, we had 29,215 detections on wildland areas in 10 years of data. The 5 km grid size and the 20 points threshold for the algorithm clusterization process was heuristically set, as the author's team evaluated several values.The coordination system is Pseudo-Mercator but the calculation of the distance corrects for the inherent distortion of the projection due to the latitude. Thus, distances were measured in meters.**

153-159: Missing references for these methodologies. (Also, a diagram would have been helpful.)

**Thanks for the commentary. We added a new figure with an explanatory diagram.**

163: Whenever the word heuristically is used, there should be a justification of the heuristics being applied and ideally an estimate of the uncertainty involved.

**The work has been edited as this point you're mentioning was very important. We've done an exploratory analysis of algorithm outputs among different alpha values from 1 km to 10km. We have looked for every polygon generated for each**

**level and concluded that lower alpha values tend to generate smaller and splitted polygons for each fire. From this point we noticed that the previously used alpha value (1 km) underestimated the burned area and so higher values were better for our purpose. Lastly, after an extensive revision, we concluded that the best alpha value was 10 km.**

Figure 2: The labels a) and b) are not clearly applied. The yellow points appear in b) only. There are no yellow polygons. The VIIRS fire detections at such high resolution should not be visualized as points as they are at least 375 x 375 m in extent.

**We have improved the caption following your suggestions. The hotspots are usually represented by points for improved visualization. We have added the VIIRS spatial resolution in the figure caption.**

175/176: There is no description of the final step of the algorithm, that is, the selection of onefinal vector. Is it the one of maximum length, some sort of average, a Gaussian model? What drove the choice of method, and what is the variability of the outcome? It seems to me that each ROS value should come with an uncertainty. As fire can grow in complex ways between successive detections, there is a need to report on what was found - and given the dataset was only 254 fires, case studies should be presented that show typical cases beyond Fig. 2 only.

**Our idea is to analyze the ROS in the head of the fires given that it is the more important metric for fire agencies and less impacted by suppression resources. Therefore, we selected the vector with the maximum ROS by time step. We have better shown this in the new Figure 2.**

179: These are not false detections. They are true detections of thermal anomalies that are not of interest to the study.

**Thank you for making this point. We have clarified our meaning in the text.**

193-195: This sentence is unclear to me.

**We have added a new Figure 2 with a self explanatory diagram of this process. Some minor changes were done for this paragraph too.**

203-208: Section 3.1 should be expanded as a lot of questions remain open. These 254 fire events led to a substantially higher number of "fire spread timesteps". From Fig. 5 my guess is that their number was 758 or thereabouts. Did each of the 254 events contain at least one spread timestep? (If yes, that would be almost surprising - was there anything in the clustering methodology to ensure that each fire had at least two successive acquisitions?) How were the fire spread events distributed - my guess is that a small number of long-running fires dominate the fire spread events. A histogram would be helpful. Also, I miss a discussion of latitude effects on the likelihood of repeat fire detections (because of satellite orbital properties). The entire discussion in 3.2 and 3.3 is tainted if these biases aren't transparently described first.

**We have adressed your question and that paragraph was changed in order to explain this process better. We need a minimum of 2 time steps to have ROS vectors in a fire. We haven't assessed latitudinal variations as an error source as there was no substantial difference within our study area (6⁰ degrees approximately)**

Figure 3: The caption, and the preceding text, should make clear that this burnt area is not the same as that detected in remotely sensed burnt area products, or delineated in the GIS systems maintained by fire managers.

**We totally agree with this comment. We have put this information in the caption.**

Figure 4: Fires were not detected contrary to the datasets made available by fire mangagement agencies (eg. https://www.ble.de/DE/BZL/Daten-Berichte/Wald/wald_node.html ) . This is understandable but needs to be discussed.

**Addressed. Please, see comment about the delineation of the study areas.**

251 ff (3.4 and Fig. 5): The values of n vary wildly between the classes. So maybe classes could be grouped to generate similarly sized datasets. What do the error bars represent?

**Our objective was to evaluate the same classes generated in the Corine Land Cover product to avoid subjectivity in the decision process of grouping land cover types. To ensure the representative of land cover type classes for further analysis, we created the category "others" to include those land cover type groups with with less than 10 observations (Figure 5).**

**Error bars represent the standard error calculated as follows:**

$$\sigma_{\bar{x}} = \frac{\sigma}{\sqrt{n}}$$

269: The authors should agree on one choice of spelling of burnt/burned area.

**Addressed. Burned area.**

270 ff: The authors discuss some sources of biases (fire size), but should expand on the shortness of their dataset (only 9 years of fires) and how it can skew the results regarding fire activity timing.

**We have indicated that there were initial 326,935 fire detections (29,215 filtered for wildland areas) and they were evenly distributed in our study area. Hence we can infer there was not much spatial bias. 9 years of fires could be a limited time range but this work relates to the usage of VIIRS dataset and we used the entire record. In this new manuscript we have discussed about biases that comes with using this dataset : spatiotemporal resolution and minimum threshold of 20 points for ROS vectors generation**

286 ff: Not all fire management areas apply the same prescribed burn processes, and permitting is also not homogeneous. This paragraph should be shortened and moved to an earlier location in the manuscript, as it addresses a very minor point. There are some formatting issues with parentheses.

**Thank you for this comment. We have removed this paragraph and added a short sentence in the previous paragraph to confirm that the detections are unlikely to include prescribed burns.**

300/301: What sensor limitations?

**Addressed. Spatio-temporal limitations of data.**

341/345: The VIIRS 375 m is unlikely to detect any smoldering peat fires, so this is not surprising at all. You correctly state that what you're seeing is surface fires. The spread of the peat fire is entirely invisible to your methodology.

**Agreed. We have removed this mention of smoldering fire.**

347: Formatting issue.

**Addressed.**

366ff: How are the low-to-moderate values you're getting impacted by averaging over a diurnal activity variation? Actual instantaneous spread may have been much faster.

**We agree with the reviewer but, unfortunately, we can not have a satellite with a shorter time overpass. For instance, we may expect higher ROS if we would have the progression of wildfires hourly.**

401 ff: Please remove redundancy in the Conclusions section with what already has been said.

**Addressed**

To conclude, in my view a quick, accurate and well-understood method to calculate VIIRS-based ROS for fire events would constitute a valuable and welcome contribution to the scientific record and toolset at the disposal of the fire management community. But the authors need to be careful to clearly describe the statistical limitations of their approach when it comes to statements about the NW-European fire regimes, and expand the presentation of the methodology itself.

**Thanks for the comments. The work has been edited considering all comments raised by both reviewers that were very useful to improve our manuscript, including the methodology, results and discussion.**

---

## Author Response (AR2)

**Reviewer 1**

The manuscript "Characterizing the Rate of Spread of Wildfires in Emerging Fire Environments of Northwestern Europe" by Mario Tapia et al. was resubmitted in revised form. The manuscript is substantially improved from its original form, specifically:

- Fig. 2 & 4 are much improved, and Fig. 5 is now believable. Clearly some additional data analysis was performed, which removed the bulk of the inconsistencies
- The exclusion of some fires in France and Germany also solidifies the analysis as the fire regime under study here is now more credible as a coherent biogeographic area. As the authors decided to retain the by-country analysis, this was really necessary. (I still don't see this analysis as overly useful, but at least in the current form it is reasonably coherent.)
- The description of clustering and filtering makes sense to me now, as does the description of the algorithm.
- Overall, the manuscript reads much better, and the small points I raised have been addressed.

**Thank you very much for all the comments. They were very useful to improve our manuscript.**

There remain some weaknesses, but I think I can understand where they come from. With only 327 vectors from a bit more than 100 fire events, providing an error estimate for the ROS may not be very meaningful. I think the authors should address this clearly in the discussion section as a limitation, especially given that the selected example in Fig. 2 does have enough vectors to allow for calculation of a standard deviation for example.

**We clearly addressed this limitation (lack of data) in the discussion as suggested. Note that we are not estimating variation of ROS within each fire so we did not analyze the standard deviation by fire (Fig 2). The descriptive analysis of the vectors was performed on the factor "Month" and "Country", in this case there was no associated error estimation due to lack of data. However, we did show the variability of ROS by factor through box plots, showing range and quartile as a deviation measurement as well. (Fig 4).. We did analyze the standard error on the "Land Cover" factor, indicating the number of observations per group.**

The other point I would ask the authors to elaborate on in 1-3 sentences is the sentence (l. 141 f.) "Cell size and 20 hotspot threshold values were heuristically set." What happens when the cell size is much smaller/larger? Also, how many vectors can be obtained typically from 20 hotspots? (Maybe this should go into the results section.)

**When cell size is larger, there is a higher probability that an "active fire cell" (i.e., cells where a fire hotspot is located) has a neighboring active cell. Thus, hotspots located in these cells will be merged into the same fire cluster. When cell size is smaller, it is more likely that these two active fire cells will be clustered into different**

**fires. We heuristically defined the cell size as a minimum distance in which two co-occurring hotspot detections can be precisely distinguished as two different fires.**

**The number of vectors basically depends on fire duration, number of time steps and, thus, the number of vertices generated for each fire.**

There are also some minor infelicities of language & punctuation that have crept back in during revision, which should be addressed through careful proofreading at the copy editing stage.

**We have corrected these faults in the new revision of the article.**

My overall judgment is that automatically deriving ROS from VIIRS will surely be well received in both the user community and the study of fire regimes, and while the work as presented leaves some avenues for future refinement and research, it is a respectable and worthwhile contribution to the research area, ultimately worthy of publication.

**Thank you very much for your comments.**

**Reviewer 2**

Summary: The authors present a new dataset of the rate of spread for 102 fires in northwestern Europe extracted from VIIRS thermal anomalies. They use this dataset to explore the seasonal pattern and differences between vegetation types of ROS. The manuscript is overall well written and has a logical flow to it. While it is great that the authors have produced this new dataset, my major comment on the manuscript is that the analysis performed seems a bit superficial at moments.

Major comments:

I find it a bit of a weird paper in the sense that the manuscript reads as if the authors extracted these new ROS data for a set of fires, and then didn't really know what to do with it. Specifically, I miss a clear research question, with the current manuscript only giving a relatively simple description of the RoS values and some comparison between seasons and vegetation types.

**We stated that the goal of this study was to characterize the ROS in an area of increasing risk in Europe in the last sentence of the introduction. We have quantified and described the ROS and its variability in a territory that has not been as well studied as, for instance, the Mediterranean regions. So, we created this novel dataset for the scientific community and a replicable methodology that could potentially be used in the future with improved input data. With this, we analyzed the temporal variation of ROS, variation among land covers and also for the final burned area, separately, as an initial but necessary characterization. We think our findings clearly fit in the scope of the journal since they addressed the aforementioned research questions.**

On the other hand, for a paper describing a new dataset, the dataset seems too limited in scope, only covering 102 fires, almost all of which occurred in the UK.

**We analyzed the entire VIIRS dataset since the beginning of satellite data acquisition in 2012. The study area was defined *a priori*, in order to develop our understanding in this understudied and increasing fire risk area: NW Europe. Thus, the boundary for this was the Northern Atlantic biogeographical region above 49th parallel, as is already justified in Materials and Methods. We agree that the dataset is limited, but the number of clustered fires and their spatial distribution is a result of the proposed methodology and is part of the characterization itself. Indeed, this speaks to the necessity of characterizing wildfire behavior in these regions of increasing risk, in lieu of pre-existing comprehensive datasets.**

Other papers have been published this year describing similar methods of extracting fire behavior information from VIIRS active fires (of which I am a co-author for all clarity), see references below. I don't know whether the current paper was inspired by previous

presentations on the subject, or whether the methods were developed in parallel. Be it one way or another, in my opinion, the novelty of this paper should lay in the science questions you want to answer with the new RoS data produced, which is currently an underdeveloped part of the paper. For example, if the focus of the paper is on increasing climate-driven fire risk in NW-Europe (which I deduce from the introduction), I would expect to see some analysis in this regard.

**Thank you for your comment, it has been very interesting to revisit your articles, which are already cited in the new revised version of the manuscript. For this article, we are interested in characterizing ROS in an area of increasing risk with a novel methodology. Studying the climatic drivers of ROS is outside the focus of this article.**

The authors make relatively bold statements on differences between countries e.g. Line 229 " Fires observed for Germany, The Netherlands and Northern Ireland appear to deviate from this pattern.". Such statements are based on data from 1 or 2 fires in the case of Germany and the Netherlands, and therefore most probably not robust. I would like to see a more robust statistical analysis to avoid these kinds of statements.

**We agree with the reviewer in the sense that the lack of data does not allow us to find differences in terms of ROS among regions. We recognize our data can not be used to perform hypothesis testing or further analysis over ROS on this. Thus, we want to emphasize that we are not doing this with "Region" as a factor, as we did with "Land cover". Taking this in consideration, we consider that it is still important to consider the few observations referring to these countries, since a low number of fires is also a descriptor of the fire dynamics in these countries.**

Minor comments:

Line 21: There is a point too much after Median.

**Addressed**

Line 114: you indicate that you used the NRT product, but I guess this is a confusion and only the case for the last couple of months of the data you used, as this product is removed after 2-3 months, once the standard science quality data is available.

**Addressed**

Figure 1: So, in practice, the dataset on RoS you generated covers the UK and Ireland, with barely any fires in the Netherlands and Germany, and none in France or Denmark. I have to agree with the previous review that this selection of the study region is somewhat odd, and I don't think that you can say that these results are valid for the mainland Europe part of the study region, as these don't seem to be sufficiently represented in the dataset.

**As indicated above, the study area was justified in the Materials and Methods section. We wanted to represent NW Europe as the Atlantic bioregion above the 49th parallel, since this is an underrepresented area in wildfire science. This criterion was established a priori, so we sought all available information in the VIIRS historical record. Although there are fewer fires in some regions than in others, this reflects the fire regime of each region by itself. On the other hand, we have not stated in the article that we are fully representing mainland Europe. Mainland Europe encompasses different bioregions outside the scope of this paper, with probably different fire regimes.**

Line 181: I am a bit surprised by the fact that you calculate ROS by connecting the new vertex to the closest previous vertex, and not by calculating the minimum distance from the new vertex to the previous perimeter (as this is your best estimate of where the fire line was). The authors have realized this, as they decided to add extra vertices (line 184), so why not go for the more straightforward option of directly calculating the minim distance to the perimeter?

**Yes, very interesting option. We agree that it would have been a better approach to compute the distance as it is suggested. The main reason not to do it in the present code is that it would make the overall workflow a bit more complex as it would require to keep track of new points that are not simply the middle point of each section, but most importantly, because we did not think about that possibility during the development. For future analysis we will definitely try to include this improvement in the algorithm.**

Line 224-225: RoS is calculated at 12h timesteps. However, the detection probability in VIIRS is much higher at night than during the day. Do you know how this might have influenced your results?

**Considering that perimeter generation relies on edge hotpots, there should be a negligible magnitude of error associated with this phenomenon, as it can only influence ROS and Burned Area estimations if there is a hotspot that is being missed by this difference in detection probability and, at the same time, this point is also part of the perimeter construction (located at the edge of the fire). Thus, this effect could be more likely to happen in smaller fires. If this happens, the effect on our results may be different depending on the variable. There could be an underestimation for burned area by day due to the omission of a hotspot during the perimeter generation. On the other hand, there could be an overestimation of the rate of spread from day to night, as we are skipping a hotspot that should have been made part of the previous perimeter and consequently considering a larger distance over a shorter time.**

Figure 4: I find it a bit odd to see the y-axis label on the left side of the figure, while the values are on the right-hand side of the figure, and would suggest putting both on the lefthand side. Also, but this is a bit more personal preference, I generally prefer not to include the grid as a background within each plot.

**Thank you for this comment. This specific edit has been made for the new revised version of the manuscript.**

Figure 5: How was the land cover type calculated here? Is it the dominant type within a fire, or has this been calculated at a 12h basis with each new RoS estimate?

**Information associated with land cover was extracted for each vector in its respective overpass.**

Figure 6: Did you take the mean, median, or max RoS per fire here? This is a nice finding, but also here, a deeper exploration of what is driving fire size could make the paper more interesting. Does RoS explain much more of the final fire size compared to fire duration, landscape type, burnable surface (as these are often patchy burnable lands), etc.?

**The vector data used for this model was the maximum ROS per fire and time step. We have specified this in the methods and results.**

Best Regards,
Stijn

References:
Hantson, S., N. Andela, M. L. Goulden and J. T. Randerson (2022). "Human-ignited fires result in more extreme fire behavior and ecosystem impacts." Nature Communications 13(1): 2717.

Chen, Y., S. Hantson, N. Andela, S. R. Coffield, C. A. Graff, D. C. Morton, L. E. Ott, E. Foufoula-Georgiou, P. Smyth, M. L. Goulden and J. T. Randerson (2022). "California wildfire spread derived using VIIRS satellite observations and an object-based tracking system." Scientific Data 9(1): 249.

**Editor comments**

From my own reading, and from reading the reviewer reports, it appears that your manuscript has significantly improved. However, several issues still remain. These will have to be addressed before publication of your manuscript can further be considered. I list here some of the remaining issues:

- I concur with the reviewers that the study area delineation remains somewhat cumbersome, especially now that from looking at Figure 1 only a handful of fires on the European mainland seems to be included. The sample size for these regions/countries seems critically low. Can you consider limiting your study to the British Isles?

The inclusion of Germany and the Netherlands in Figure 4 seems problematic. Can you please justify this, or alternatively remove this regions from the study?

**As mentioned above in the responses to reviewer 2, we chose to study Northwest Europe as the Atlantic bioregion above the 49th parallel because it is a region underrepresented in wildfire science. To develop a novel method and to study this region was our main objective. For this we sought all available VIIRS historical records. In addition, we want to emphasize that we are not doing hypothesis testing or deeper types of statistical analyses over "Country" as a factor, as we acknowledge that there is no sufficient sampling size for each one. At the country level, we want to do general analyses of ROS for all available data. Although some regions have fewer fires than others, this also reflects the fire regime of each region itself.**

- I checked your supplementary data files and wonder why you have not chosen for a data format like shapefiles. Now, there are separate files for the geospatial data (KML/Z files) and its attributes (CSV files). I believe this dataset could be more elegantly shared using for example a shapefile format. In addition, while it is clear to me what the added value of the 'point' and 'polygon' layers is, it is not entirely clear to me what the added values is of the 'vector' and 'raster' layers. Can you please clarify? Including a readme file with your dataset could also be helpful.

**The algorithm works with kml files as output. Files have internal temporal metadata meant for visualization in, for instance, Google Earth. Conversion to shapefile loses information for this. Then, we propose to gather all the separate KML files into a smaller number of grouped files, for a more adequate presentation of the results.**

**CSV files contain explicit Rate of Spread information. We added this type of data for a faster analysis of the data, with their respective coordinates, initial and final time stamps.**

- Some new papers have been published covering related methodologies. I recommend a discussion of the similarities and differences of your methods with these published methods. Here are some references that you could consider including:

Hantson, S., N. Andela, M. L. Goulden and J. T. Randerson (2022). "Human-ignited fires result in more extreme fire behavior and ecosystem impacts." Nature Communications 13(1): 2717.

Chen, Y., S. Hantson, N. Andela, S. R. Coffield, C. A. Graff, D. C. Morton, L. E. Ott, E. Foufoula-Georgiou, P. Smyth, M. L. Goulden and J. T. Randerson (2022). "California wildfire spread derived using VIIRS satellite observations and an object-based tracking system." Scientific Data 9(1): 249.

**As mentioned for Reviewer 2, we are considering this for the revised version of the manuscript**

- Please consider a sensitivity analysis of the threshold of 20 fire pixels for considering a fire cluster.

**Indeed we consider that it is important to review the result of the algorithm and the perimeters as a function of the parameters introduced. One of the first steps within this work was to review all the perimeters generated as a function of different values of α. As we explained in Materials and Methods, we assessed 4 values of α: 1, 3, 5, 10 until we found shaped polygons spatially coherent with their shape and with the position of the hotspots and so we finally fixed 10 as the most coherent value for fire perimeters.**

---

## Author Response (AR3)

We appreciate each and every one of your remarks. We have emphasized our research's primary focus, which is on major fires in NW Europe. In order to show what would be our lowest threshold that our algorithm works with, we also provided a description of the smallest fire detection in our log. We've taken into account your request for a title change. As suggested by the editor, the title has been modified. In addition, the order of authors has ben rearranged, Tomás Quiñones has been relocated to position 4.

We are pleased with the work and all the suggestions made by the editor and the reviewers.

Best regards

Adrián Cardil, Victor Tapia, Santiago Monedero, Tomás Quiñones, Kerryn Little, Cathelijne Stoof, Joaquín Ramírez, and Sergio de Miguel